# Diversity and Cytogenomic Characterization of Wild Carrots in the Macaronesian Islands

**DOI:** 10.3390/plants10091954

**Published:** 2021-09-18

**Authors:** Guilherme Roxo, Mónica Moura, Pedro Talhinhas, José Carlos Costa, Luís Silva, Raquel Vasconcelos, Miguel Menezes de Sequeira, Maria Manuel Romeiras

**Affiliations:** 1Linking Landscape, Environment, Agriculture and Food (LEAF), Instituto Superior de Agronomia (ISA), Universidade de Lisboa, Tapada da Ajuda, 1340-017 Lisbon, Portugal; roxogguilherme@gmail.com (G.R.); ptalhinhas@isa.ulisboa.pt (P.T.); jccosta@isa.ulisboa.pt (J.C.C.); 2CIBIO, Centro de Investigação em Biodiversidade e Recursos Genéticos, InBIO Laboratório Associado, Universidade do Porto, 4485-661 Vairão, Portugal; raquel.vasconcelos@cibio.up.pt; 3CIBIO, Centro de Investigação em Biodiversidade e Recursos Genéticos, CIBIO-Azores, Departamento de Biologia, Universidade dos Açores, Rua Mãe de Deus 58, Apartado 1422, 9501-801 Ponta Delgada, Portugal; monica.mt.moura@uac.pt (M.M.); luis.fd.silva@uac.pt (L.S.); sequeira@uma.pt (M.M.d.S.); 4Madeira Botanical Group, Faculty of Life Sciences, University of Madeira, 9020-105 Funchal, Portugal; 5Centre for Ecology, Evolution and Environmental Changes (cE3c), Faculdade de Ciências, Universidade de Lisboa, Campo Grande, 1749-016 Lisbon, Portugal

**Keywords:** insular ecosystems, endemic species, Apiaceae, Daucinae subtribe, DNA C-values, morphological traits

## Abstract

The Macaronesian islands constitute an enormous reservoir of genetic variation of wild carrots (subtribe Daucinae; Apiaceae), including 10 endemic species, but an accurate understanding of the diversification processes within these islands is still lacking. We conducted a review of the morphology, ecology, and conservation status of the Daucinae species and, on the basis of a comprehensive dataset, we estimated the genome size variation for 16 taxa (around 320 samples) occurring in different habitats across the Macaronesian islands in comparison to mainland specimens. Results showed that taxa with larger genomes (e.g., *Daucus crinitus*: 2.544 pg) were generally found in mainland regions, while the insular endemic taxa from Azores and Cabo Verde have smaller genomes. *Melanoselinum decipiens* and *Monizia edulis*, both endemic to Madeira Island, showed intermediate values. Positive correlations were found between mean genome size and some morphological traits (e.g., spiny or winged fruits) and also with habit (herbaceous or woody). Despite the great morphological variation found within the Cabo Verde endemic species, the 2C-values obtained were quite homogeneous between these taxa and the subspecies of *Daucus carota*, supporting the close relationship among these taxa. Overall, this study improved the global knowledge of DNA content for Macaronesian endemics and shed light into the mechanisms underpinning diversity patterns of wild carrots in the western Mediterranean region.

## 1. Introduction

A comprehensive understanding of plant genomes is still largely missing, due to the relatively small number of species examined to date (around 12,273 species) [1]. High-throughput flow cytometry (FCM) screens for DNA content have especially aided in identifying the frequency of polyploids [2], as well as their phenotypic and geographical distributions [3]. Advances in flow cytometry are being applied to study intraspecific and interspecific variation (e.g., [4]), as well as to resolve complex low-level taxonomies, including the delimitation of species boundaries [5]. Both polyploidy and hybridization have been recognized as two of the most important sources of diversification in oceanic islands [6,7]. Suda et al. [8,9] analyzed 40% of the Canary endemic flora, revealing a low incidence of polyploid plants in this region (33.7%). On the other hand, hybridization appears to be common in most species-rich and rapidly diversifying groups [10]; in fact, some studies indicate that after multi-introductions, hybridization plays a greater role in diversification than the progressive accumulation of genetic variation through time [11]. In addition, the number of endemic species per genus was found to be negatively correlated with genus-average genome size regarding insular lineages that radiated (i.e., cladogenesis) on the Hawaii and Marquesas archipelagos [12].

The northeastern Atlantic Macaronesia Region (i.e., Azores, Madeira, Selvagens, Canary, and Cabo Verde archipelagos) shows high levels of plant endemism (around 900 endemic taxa), with several range-restricted species occurring in different habitats [13]. These islands have a remarkable genetic diversity [14] regarding crop wild relatives (CWR), which are related to globally important crops [15,16]. In Apiaceae, the Daucinae subtribe has an important economic value [17] since it includes the widely consumed carrot (*Daucus carota* subsp. *sativus*). Regarding its CWR, Macaronesia harbors 10 Macaronesian endemic taxa from five different genera, the Azorean *Daucus carota* subsp. *azoricus*; *Melanoselinum decipiens* and *Monizia edulis* from Madeira; *Tornabenea*, the only endemic genus from Cabo Verde, which radiated in six different taxa; and *Tornabenea annua*, *Tornabenea bischoffii*, *Tornabenea humilis*, *Tornabenea insularis*, *Tornabenea ribeirensis*, *Tornabenea tenuissima* (see Figure 1) and *Cryptotaenia elegans* from the Canary Islands.

Daucus carota subsp. azoricus was described by Franco in 1971 [18]. Due to infra-specific phenotypical diversity, there is uncertainty on distribution and diagnostic characters. A recent genomic study did not support the Azorean Daucus carota as a distinct taxon [19], and Frankiewicz et al. [20] suggested that the plants were dispersed by settlers. A molecular genetics study including a thorough archipelago-wide sampling is currently underway to better clarify the taxonomy of the Azorean Daucus (Moura, personal comm.).

The Melanoselinum genus was described by Hoffmann [21] to accommodate a specimen of unknown origin growing in the botanical garden of Hannover. In 1829, Richard Lowe found it along the Ribeira de São Jorge, Madeira Island [22]. Since then, various authors have given accounts of the diversity and taxonomy of this genus, one that is currently accepted as a monotypic genus [23,24,25].

In 1855, Richard Lowe found *Monizia edulis* plants growing in a sea cliff on the east side of Deserta Grande (Madeira archipelago). In 1856, Lowe [26] described this new taxon and reported it [27] in the Selvagens archipelago (Selvagem Grande) on the basis of observations by Constantino Cabral de Noronha. No further reports occurred, and Jardim and Menezes de Sequeira [25] considered it extinct in Selvagens. It also occurs on Porto Santo (Madeira archipelago), restricted to Ilhéu de Cima [28].

Undescribed specimens of *Tornabenea* were first collected by Vogel in 1841. In 1849, Parlatore, with material from a single species (*Tornabenea insularis*), established a new tribe with a monospecific genus, *Tetrapleura* [29]. The name was a homonym and renamed to *Tornabenea* by Parlatore [30]. In 1851, J. A. Schmidt collected material later described as *Tornabenea bischoffii* and *Tornabenea hirta* [31]. These were considered synonyms by Lobin and Zizka [32], while Martins [33] included both taxa in the “*Cabo Verde Flora*” as synonyms of *Tornabenea insularis*. The number of accepted species in the genus has recently varied from three by Martins [33] (*Tornabenea annua*, *Tornabenea insularis*, and *Tornabenea tenuissima*), to five by Brochmann et al. [34], who included Tornabenea bischoffii and Tornabenea humilis, to six by Sánchez-Pinto et al. [35], who included *Tornabenea ribeirensis*.

Despite the publication of several phylogenetic studies to resolve the evolutionary relationships among the major clades of Daucinae subtribe (e.g., [36,37,38,39,40]), some nodes in the Daucinae tree have proved to be extremely problematic and have remained unresolved, particularly among the Macaronesian endemic taxa. Recent studies including the analysis of wood anatomy, life form, life span, and reproductive strategies led to a reappraisal of the endemic genera *Melanoselinum, Monizia*, and *Tornabenea* and their inclusion in the *Daucus* genus [20,41,42]. Moreover, using genotyping-by-sequencing, Martínez-Flores et al. [43] provided evidence that at least some *Tornabenea* species should be included within the *Daucus carota* complex, constituting a remarkable case of allopatric speciation in the Cabo Verde Islands with total or rarely almost total reduction of mericarp spines. However, only two species (i.e., *Tornabenea annua* and *Tornabenea tenuissima*) were evaluated by the authors. Therefore, relationships within Macaronesian Daucinae taxa are still poorly understood and a good field-sampling is needed, along with fossil data to produce a robust phylogeny [44]. Nevertheless, Góis-Marques et al. [45] recently described the oldest macrofossil of a wild carrot and the first fossil of a plant displaying insular woodiness (i.e., *Melanoselinum decipiens*), providing a calibration for future Apiaceae phylogenies.

The low resolution of the phylogeny within the Macaronesian group outlined the difficulty of an accurate comprehension of the systematics and the relationships within the Daucinae subtribe. Thus, unraveling the variations in genome size (also known as C-value) within this subtribe can be highly relevant to understand its diversification in the Macaronesian region. Although the meaning of the variation in the C-value is still not completely understood, it is known that the variation can be correlated to non-coding DNA, more specifically transposable elements [46]. Several life-history traits have been found to correlate with genome size through the associated effects of nuclear DNA content on cell size [47]. Similarly, significant associations between genome size and conservation status suggest that threatened species are more prone to be associated with larger genomes [48].

Native plants from Macaronesian islands are significantly underrepresented in the existing genome size database [1] but show a remarkable range of genome size [9]. Regarding the endemic taxa of the Daucinae subtribe, only two species have been analyzed thus far, namely, *Daucus carota* subsp. *azoricus* (2C-values = 1.64 ± 0.02 pg [49]) and *Cryptotaenia elegans* (2C-values = 0.94 ± 0.01 pg [9]). Moreover, the counting of chromosomes has also been used extensively as an important phylogenetic character in the context of cytotaxonomy [50], helping in the definition of different taxa.

Within the Macaronesian islands, molecular studies with CWR have been focused on establishing their taxonomic status and depicting phylogenetic relationships, e.g., [15,51]. However, few studies have been carried out using cytogenomic data, complemented with data on the morphology and ecology of CWR species. The present study aimed to review of the morphology, ecology (including associated plant communities), and conservation status of the Daucinae species, and on the basis of a comprehensive dataset, we estimated the genome size variation for 16 taxa, which occur in different habitats across the Macaronesian islands in comparison to mainland specimens. Such data are needed because most of these CWR are endemic species within these island ecosystems, and their presence and conservation are likely to be threatened by ongoing climate change.

## 2. Results

### 2.1. Reappraisal of Morphology, Ecology, and Conservation Status of Daucinae in Macaronesia

A total of 16 taxa were sampled (Table 1), nine of them with endemic status: one from the Azores (*Daucus carota* subsp. *azoricus*), two from Madeira (*Melanoselinum decipiens* and *Monizia edulis*), five from Cabo Verde (*Tornabenea annua, Tornabenea bischoffii, Tornabenea insularis, Tornabenea ribeirensis,* and *Tornabenea tenuissima*), and one from mainland Portugal (*Daucus carota* subsp. *halophilus*) (see Figure 1).

Most of the endemic species sampled in the Macaronesia are single-island endemics except *Daucus carota* subsp. *azoricus*, which occurs in all the Azores Islands; *Monizia edulis*, found in all of the Madeira sub-archipelagos; and *Tornabenea insularis*, the only *Tornabenea* species occurring in more than one island of Cabo Verde (Brava, São Nicolau, and São Vicente).

Regarding the ecological data (Table 1), *Daucus carota* subsp. *azoricus* is a coastal taxon that occurs in sea rocks, cliffs, pastures, uncultivated or cultivated lands, and roadsides. *Melanoselinum decipiens* and *Monizia edulis*, both endemic species from Madeira, occupy different ecological zones, that is, shady and rocky places of clearances of the laurel forest and fissures in cliffs, respectively. *Tornabenea* species occupy similar ecological zones on Cabo Verde islands, being five of the six species single-island endemics (the exception being *Tornabenea insularis*). Only *Tornabenea humilis* is found in lowland/coastal areas below 500 m altitude at Fogo, whereas the other five species are restricted to sub-humid habitats of the mountain areas above 600 m.

The conservation status of the sampled species revealed that six of them (38%) are threatened: three (20%) are critically endangered and the other three (20%) are endangered (Table 1). Among the threatened species, five occur in Cabo Verde (i.e., *Tornabenea* genus) and the sixth, *Monizia edulis*, occurs in Madeira. The archipelago of Cabo Verde is the one presenting the highest number of threatened species of the Daucinae subtribe.

The Daucinae subtribe is mostly composed of hemicryptophyte taxa, namely, *Daucus carota* subsp. *azoricus*, *Daucus carota* subsp. *carota*, *Daucus carota* subsp. *gummifer*, *Daucus carota* subsp. *halophilus*, *Daucus carota* subsp. *maximus*, *Daucus carota* subsp. *sativus*, *Daucus crinitus, Tornabenea annua, Tornabenea insularis,* and *Tornabenea ribeirensis.* Moreover, the chamaephyte habit is only seen in the endemic Macaronesian species (Table 1). Similarly, the winged secondary ribs are traits only present in the endemic species of the Macaronesia (Table 2).

In the Daucinae subtribe, the morphological characters such as the leaves, fruits, and umbels are usually used to recognize the infratribal taxa [18]. For the identification of some subspecies of *Daucus carota*, the degree of contraction in the fruiting umbels is a key trait; the presence and absence of a sterile flower in the center of the umbel is also a character that helps to distinguish between *Daucus carota s.l.* and *Daucus muricatus*. Moreover, the leaves of *Daucus crinitus* are very relevant for the identification of this taxon, as they exhibit a sessile or subsessile segment that is not observed in the other *Daucus* species (see the Materials and Methods section). Traits such as bracts, bracteoles, and fruits are also very useful to distinguish genera, namely, between the genus *Daucus* and the related endemic Macaronesian genera (i.e., *Melanoselinum, Monizia*, and *Tornabenea)*. *Daucus* is characterized by fruits with spiny secondary ribs and pinnate bracts. In contrast, *Melanoselinum* taxa exhibit winged secondary ribs that are serrated and irregularly cut bracts. *Monizia* fruits have swollen and corky ribs, as well as bracts and bracteoles with fringed margins (see Table 2). The genus *Tornabenea* is more similar to *Daucus*, but fruits are compressed dorsally in the former. Some taxa also present winged secondary ribs (*Tornabenea bischoffii* and *Tornabenea ribeirensis*), opposed to the spiny secondary ribs in *Daucus.* Still, the *Tornabenea* taxa that presented spiny secondary ribs have spines that are less differentiated from the spines of *Daucus* (see Figure 1).

### 2.2. Cytogenomic Characterization

The cytogenomic results obtained are summarized in Table 3, which shows the mean 2C-values in picograms (pg) with standard deviations (SD), coefficient of variation (%) for the 16 analyzed species, and 2C-values estimates evaluated by Nowicka et al. [49]. Genome size determinations based on flow cytometry produced histograms of fluorescence of G0/G1 peaks (Figure 2), and the coefficient of variation (CV) values ranged from 2.31% to 5.99% (mean 4.12%) for the analyzed specimens (Table 3).

Estimations for 16 species within the Daucinae subtribe are presented (Table 3), of which 8 had already been previously estimated by Nowicka et al. [49]; the values obtained in both studies were very similar, with average differences of 11%, ranging between 26% for *Daucus carota* subsp. *maximus* and 4% for *Daucus carota* subsp. *halophilus*. The genome size of the analyzed species was quite homogenous (1.349 ± 0.442 pg), particularly in *Tornabenea* genus (1.235 ± 0.016 pg), ranging from 1.214 ± 0.026 pg in *Tornabenea ribeirensis* to 1.259 ± 0.211 pg in *Tornabenea bischoffii*. Values were the smallest for *Pseudorlaya pumila* (Table 3). Moreover, no differences were found between the *Tornabenea* specimens and *Daucus carota* specimens, with the exception for subspecies *carota*. *Melanoselinum decipiens* and *Monizia edulis*, both from Madeira, showed intermediate values, also differing significantly from all the remaining taxa. *Daucus crinitus* and *Daucus muricatus* specimens from mainland Portugal showed the highest values, which were significantly different from those for all the remaining taxa (Figure 3).

The results of the application of different GLMs (Table 4) showed that the most explanatory factor was the taxon, with the lowest Akaike’s information criterion (AIC) and highest adjusted R^2^, which was also confirmed using the Kruskal–Wallis test (chi-squared = 29.061, df = 3, *p*-value = 2.174 × 10^−6^).

The genome size variation for the studied Daucinae taxa according to fruit secondary ribs (Figure 4A), habit (Figure 4B), Raunkiaer classification [52] (Figure 4C, see Appendix A), and regions (Figure 4D) is presented in Figure 4. With some exceptions, the genomes of herbaceous taxa were smaller than those from woody taxa, with a similar pattern between taxa with spiny and winged fruits. The exceptions are obvious in Figure 4A,B. Regarding herbaceous taxa, two mainland species appeared with very high values, therefore originating a high level of heterogeneity, justifying the low adjustment of the respective GLM (Table 4). Likewise, concerning taxa with spiny fruits, the same taxa from the mainland originated a high level of heterogeneity, justifying the low adjustment of the respective GLM (Table 4).

When considering Raunkiaer life-forms (See Appendix A), therophytes tended to show the highest values and hemicryptophytes the lowest. However, one of the therophytes showed the lowest value and one of the hemicryptophytes showed the highest, therefore originating a high level of heterogeneity (Figure 4C), justifying the low adjustment of the respective GLM (Table 4). When considering the different regions, the Azores (1.167 ± 0.006 pg) and Cabo Verde (1.238 ± 0.059 pg) showed the lowest mean values, followed by the specimens from mainland Portugal (1.531 ± 0.483 pg) and Madeira (1.617 ± 0.183 pg; Figure 4D). However, it should be noted that specimens from mainland Portugal included taxa with a wide range of values, from the lowest to the highest, as mentioned above when comparing the results by taxon. As also seen above, the two endemic taxa from Madeira showed intermediate results, while the smallest values for that archipelago were associated with specimens from *Daucus carota* subsp. *carota*. This heterogeneity justifies the low adjustment of the respective GLM (Table 4)

## 3. Discussion

The Daucinae subtribe comprises taxa of high economic importance such as *Daucus carota* subsp. *sativus* (cultivated carrot), and its CWR represent invaluable genetic resources, with genes that can improve traits such as productivity and resilience in agriculture [17]. Our study provides new cytogenomic data for eight endemic taxa from Azores, Madeira, and Cabo Verde archipelago that can contribute to understanding the relationship of the Daucinae subtribe within the Macaronesian islands, where several threatened taxa exist. Overall, with the previous estimates stored at the Angiosperm DNA C-values database [1], only one Macaronesian endemic taxon from the subtribe Daucinae is missing (*Tornabenea humillis*, endemic to Fogo Island, Cabo Verde).

In general, the species with the largest genomes exist in mainland Portugal (*Daucus crinitus* and *D. muricatus*), and the insular taxa presented lower mean 2C-values compared to the continental ones. This tendency towards small genomes in islands has been observed in Macaronesia [8,9] and in Hawaiian and Marquesas archipelagos [12]. Even though the Azores and Cabo Verde presented lower mean 2C-values than the mainland Portugal, and this could support the above-mentioned hypothesis, it is important to note that continental regions also presented the taxon with the lowest genome size (i.e., *Pseudorlaya pumila*), and Madeira presented endemic taxa with larger genome sizes than continental taxa (i.e., *Melanoselinum decipiens* and *Monizia edulis*).

According to our data, 2C-values are quite homogeneous across the *Daucus carota* complex (including *Tornabenea* species), supporting the close relationship among the taxa, previously mentioned by several authors [20,41,42,43]. Although the use of genome size as a taxonomical marker has been widely used with other plant lineages, e.g., [4,53], *Daucus carota* complex is considered one of the taxonomically most difficult groups within the Apiaceae family and our 2C-values were unable to clearly discriminate the *Daucus carota* subspecies. The high outcrossing rate (around 96%) in wild carrot implies that high frequencies of gene flow may occur among the different subspecies [54]. In addition, pollen of wild carrots could be dispersed by insects over a long distance, and cultivated and wild carrot are fully inter-fertile, often overlap in flowering time, and hybrids may sometimes have high fertility and viability. While gene flow within *Daucus carota* complex appear extensive, more work is still needed (e.g., integrative genomic and morphological studies) to clearly discriminate population structure within the *Daucus carota* complex, as well as using additional samples from more diverse geographic origins to provide future support for recognizing some species and geographically defined subspecies [19,43].

In particular, the molecular relationships between some *Tornabenea* species and *Daucus carota s.l*. have been difficult to establish, and recent phylogenetic studies [20,41,42,43] provided all evidence that at least some *Tornabenea* species should be included within the *Daucus carota* complex. As stated above, our results also revealed very little cytogenomic differentiation between *Tornabenea* and *Daucus carota* specimens, confirming the need for a clarification of the taxonomy of *Tornabenea* species, e.g., [33]. Most of the currently recognized *Tornabenea* taxa are single-island endemics or ecologically isolated species occurring on the same island, as it is the case of *Tornabenea tenuissima* that is restricted to sub-humid montane areas mainly above 1200 m on Fogo, while *Tornabenea humilis* is found in lowland/coastal areas below 500 m altitude on this island (see Table 1). Moreover, the rare *Tornabenea ribeirensis* is restricted to shady and seasonally damp valleys of very few streams in the North side of São Nicolau [55], whereas *Tornabenea insularis* is found in open, non-sheltered, and sometimes extremely exposed habitats and in a different plant communities alliance (see Appendix A). Directional selection into diverse island environments is reported to many of the well-known spectacular examples of adaptive radiation, such as *Echium* species in the Southern Cabo Verde Islands in which two lines of speciation occur (*Echium vulcanorum* and *Echium hypertropicum*), each driven by selection within markedly different ecological zones [56]. The same pattern can be seen in Madeira Island with *Melanoselinum decipiens* and *Monizia edulis*, in which both occur in a phytosociological very distinct communities (see Appendix A), with the former occurring in forest clearances and the latter in succulent-rich scrub volcanic rock substrates and walls.

The morphology of the secondary ribs might also provide some indication of the recent colonization and radiation of the *Tornabenea* genus [33,55]. The presence of winged secondary ribs in some taxa is likely to be a reversal evolution and linked to insular habitats as proposed by Wojewódzka et al. [42]. In fact, such a trait is only seen in insular taxa (*Monizia edulis, Melanoselinum decipiens, Tornabenea bischoffii*, and *Tornabenea ribeirensis*). Spiny ribs are seen as an advantageous trait for epizoochorous dispersion and might have been lost due to the general absence of native terrestrial mammals in volcanic islands, which were never connected to the mainland [42]. This loss of dispersal abilities through epizoochory is a common feature in oceanic islands [57]. On oceanic islands, there is usually a reversal evolution from spines to wings, as this feature is ideal for wind dispersal, as well as an increment of the fruit size [57]. Alternatively, this characteristic may have been lost by drift. Altogether, we can hypothesize that the island taxa with winged secondary ribs are older than the ones with spiny secondary ribs. Our results showed that taxa with spiny ribs have on average significantly smaller genomes than taxa with winged secondary ribs. The larger genome of taxa with winged secondary ribs might be related with the increase of mericarp size in insular taxa. In fact, such a positive correlation between seed mass and genome size has been shown by various authors [4,58].

Insular woodiness and a perennial life cycle are a key evolutionary innovation that drives radiations in insular systems [59]. Moreover, it is the lineages that developed secondary woodiness that diversified more in Macaronesia [60]. Interestingly, chamaephyte taxa presents the highest mean 2C-values when compared with hemicryptophytes and therophytes. Nonetheless, Beaulieu et al. [61] noted that mean genome sizes in the Fabaceae family were significantly smaller in woody than in herbaceous species and presented a smaller variation.

Overall, this study improved the global knowledge of DNA content for Macaronesian endemics and shed light into the mechanisms underpinning diversity patterns of wild carrots in the western Mediterranean region. Despite the relatively coherent results revealed by the cytogenomic analyses, on the basis of around 320 field-collected samples, further efforts should be performed to increase the plant collection, namely, of *Tornabenea humilis* from Fogo Island and of *Cryptotaenia elegans* from Canary Islands.

Although the Macaronesian Islands harbors a rich diversity of wild carrots, it was exposed that the endemic species in particular should be protected, as around 40% of the studied taxa are threatened, endemic, and red-listed species (i.e., 20% are critically endangered and 20% are endangered; see Table 1). Presently, it is widely recognized that only the conservation of these populations in their habitats (in situ conservation) will ensure the continued supply of the novel genetic material, critical for future crop improvement [62]. Thus, our results revealed that the C-values are quite homogeneous across the “*Daucus-Tornabenea-Monizia-Melanoselinum”,* indicating that these taxa can easily be used in crop improvement. Of these species, *Monizia edulis* endemic in Madeira archipelago and all the *Tornabenea* species should be prioritized taxa, and urgent conservation actions must be implemented, particularly as they have a very limited geographic range (often rare and endemic taxa) and they were classified in threatened categories (see Table 1). Finally, a better understanding of how the intraspecific diversity is changing over time and space is required, and we argue that the cytogenomic analyses can contribute additional data that can be useful to make informed decisions for the conservation of plant genetic resources in the Macaronesian islands.

## 4. Materials and Methods

### 4.1. Study Area

Taxa from the Daucinae subtribe occur in all Macaronesian archipelagos. However, our sampling focused on Azores, Madeira, and Cabo Verde, where the main genera of this subtribe occur (i.e., *Daucus* L., *Melanoselinum* Hoffm., *Monizia* Lowe., and *Tornabenea* Parl.) (Figure 5). The Azores archipelago consists of nine islands, is the northernmost of Macaronesia, and is located approximately 1300 km west of mainland Portugal. It is composed of an eastern group of islands (São Miguel and Santa Maria), a central group (Faial, São Jorge, Graciosa, Pico, and Terceira), and a western group (Flores and Corvo). The Madeira archipelago comprises Madeira, the largest island of the archipelago; Porto Santo and its six islets; and Desertas (Bugio, Deserta Grande, and Ilhéu Chão). Cabo Verde includes 10 main islands and several islets, grouped into two main sets according to the prevailing northeast winds: the windward islands, namely, Santo Antão, São Vicente, Santa Luzia, São Nicolau, Sal, and Boavista, and the Leeward islands, namely, Maio, Santiago, Fogo, and Brava.

### 4.2. Studied Macaronesian Endemics and Sampling

In Macaronesia, 10 endemic species occur, of which 8 were included in the study: *Daucus carota* L. subsp. *azoricus* Franco, *Melanoselinum decipiens*, *Monizia edulis*, *Tornabenea annua, Tornabenea bischoffii, Tornabenea insularis, Tornabenea ribeirensis,* and *Tornabenea tenuissima* (see Figure 1). The data presented in Table 1 and Table 2 were obtained from a thorough bibliographic revision, which included the protologues and a morphometric revision of the type specimens whenever deemed possible.

For cytogenomic studies, specimens of the Daucinae subtribe were collected during several field surveys in the Macaronesia archipelagos from 2017 to 2020. Fieldwork took place in four islands of Azores (Faial, Flores, Pico, and Santa Maria), two islands of Madeira (Madeira and Deserta Grande), and five islands of Cabo Verde (Santiago, Santo Antão, São Nicolau, São Vicente, and Fogo). Additionally, specimens from mainland Portugal (Beja, Faro, Leiria, Lisboa, and Setúbal) were also sampled. A total of 16 native taxa (from 48 different populations) were sampled for the study (Appendix A). Furthermore, from each population, a minimum of three specimens was collected. The samples were preserved in wet tissue paper wrapped in aluminum foil and zip-locked bags at 5 °C and posted to the laboratory; C-DNA data were obtained after a maximum period of three days.

Vouchers from the majority of the samples were collected and deposited at the João de Carvalho e Vasconcellos herbarium (LISI) of Instituto Superior de Agronomia, University of Lisbon, at the Herbarium of the University of Madeira (UMAD) and at the Herbarium Ruy Telles Palhinha, University of the Azores (AZB).

### 4.3. Data Collection

A database for the sampled species was assembled, including information for each taxon on (i) geographical distribution; (ii) accepted scientific names according to Banasik et al. [41], Wojewódzka et al. [42], and Martínez-Flores [63]; (iii) habit; (iv) habitat (herbaceous or woody); (v) conservation status; (vi) number of chromosomes; and (vii) Raunkiaer classification (Appendix A) [52].

Information on the geographical distribution (inside Macaronesia and worldwide) was extracted from the most recent checklists [25,35,64,65].

Information on the habit and morphology of all sampled taxa were obtained from floras of each archipelago [24,33,34,66,67] and included the following morphological traits: (i) leaves, (ii) inflorescence, (iii) bracts, (iv) bracteoles, (v) fruits, and (vi) secondary ribs of the fruit (spiny vs. winged).

Drawings (see details in Figure 1) were made on the basis of photographs and on herbarium specimens housed in LISI and LISC Herbaria. Ecological data including sinecological data are based on the work of Mucina et al. [68] and Rivas-Martinez et al. [69]. The chromosome numbers of the various taxa were obtained through consulting Nowicka et al. [49], Dalgaard [70], Bramwell and Murray [71], Borgen [72], Zizka [73], Grosso et al. [74], and Spooner [75]. The conservation status of the sampled species was obtained following the International Union for Conservation of Nature (IUCN) Red List of threatened species criteria [76], Corvelo [77], and Romeiras et al. [78].

### 4.4. Cytogenomic Analysis

Nuclear DNA content was estimated using flow cytometry (FCM). Preparation of suspensions of intact nuclei for analysis was performed following the method of Galbraith [79]. The fresh young leaves were chopped with a razor blade in a Petri dish containing 1 mL of woody plant buffer (WPB; 0.2 M Tris-HCl, 4 mM MgCl_2_, 1% Triton X-100, Na_2_EDTA 2 mM, NaCl 86 mM, sodium metabisulfite 20 mM, PVP-10 at 1%, pH 7.5 [80]). The nuclear suspension was sieved through a nylon mesh with 30 μm to remove large debris. The obtained nuclei were stained with 25 μg mL-1 of propidium iodide (PI; Sigma-Aldrich, St. Louis, MO, USA). For the estimation of the nuclear DNA content, the use of a reference standard of known genome size is required. The following standards were employed: solanum lycopersicum L. (2C = 1.96 pg; [81]), Raphanus sativus L. (2C = 1.11 pg; [81]), and Rhamnus alaternus L. (2C = 0.68 pg; [82]). Furthermore, the acquisition of numeric data and fluorescence graphs was made by Sysmex FloMax software v2.4d (Sysmex, Görlitz, Germany), as described by Guilengue [83]. From the analyzed samples, the diploid quantity of DNA (in pg, per nucleus) was estimated using the formula:
(1)$$\mathrm{Nuclear}\;\mathrm{DNA}\;\mathrm{Content}\;\left(\mathrm{pg}\right)=\frac{\mathrm{Sample}\;G1\;\mathrm{Peak}\;\mathrm{Mean}\;\times \mathrm{Genome}\;\mathrm{size}\;\mathrm{of}\;\mathrm{Reference}\;\mathrm{Standard}}{\mathrm{Reference}\;\mathrm{Standard}\;G1\;\mathrm{Peak}\;\mathrm{Mean}}$$


### 4.5. Statistical Analyses

Statistical analyses and descriptive statistics were performed using R software [84]. Descriptive statistics were calculated for each species, namely, mean and standard deviation (SD) of the genome size (2C-values, pg). For the data on 2C-values, descriptive analyses were performed using the boxplot statistical algorithm. Comparisons between genome size values and (i) habit (woody vs. herbaceous), (ii) Raunkiaer classification (chamaephyte vs. hemicryptophte vs. therophyte) (see Appendix A), (iii) distribution (archipelago vs. mainland), and (iv) Morphology of secondary ribs (spiny vs. winged). Group comparisons were implemented using non-parametric tests since genome size data were not Gaussian (*p* < 0.05 with the Shapiro–Wilk test [85]) even after using Box–Cox transformation [86] or other conventional transformation techniques [87]. Thus, we opted for the Mann–Whitney test for two group comparisons and the Kruskal–Wallis test for comparisons of more than two groups, using the respective functions in R. Since the Kruskal–Wallis test indicated the rejection of the null hypothesis, we applied a non-parametric multiple comparison test [88,89] using the function posthoc.kruskal.conover.test of the “The Pairwise Multiple Comparison of Mean Ranks Package (PMCMR) R package” [90], including Bonferroni-type adjustment of *p*-values, aiming to ensure a relatively high level of statistical power (i.e., reduction in the probability of committing a type II error).

We calculated different Gaussian generalized linear models (GLMs) in order to determine what factors could better explain the observed 2C-values: a Null model was used as a benchmark, a habitat model (island vs. mainland specimens); a habit model (woody vs. herbaceous); a Raunkiaer classification model (chamaephyte vs. hemicryptophte vs. therophyte); a fruit type model (specimens with spiny vs. wing secondary ribs); a full model including all the previous factors, as well as other models resulting from its simplification; and a Taxon model, comparing all the included taxa. The implementation followed Ávila et al. [91] and Parelho et al. [92] (see references therein), using the glm function of R. The best models were selected on the basis of the maximum likelihood approach using AIC [85]. The model with the lowest AIC and the highest R^2^ was considered to best fit the data. The R package “mass” was used to evaluate GLMs. Although the data failed to comply with normality, the calculated GLMs included all the samples in the analysis. Thus, according to the central limit theorem, when independent random variables are added, their properly normalized sum tends toward a normal distribution, even if the original variables themselves are not normally distributed. That is, with a large sample size (i.e., more than 100 observations in this case), the mean tends to a normal distribution, even if the underlying data are not Gaussian [93]. Since the GLMs address the mean of the distribution, we considered its application as correct in this context.

## Figures and Tables

**Figure 1 plants-10-01954-f001:**
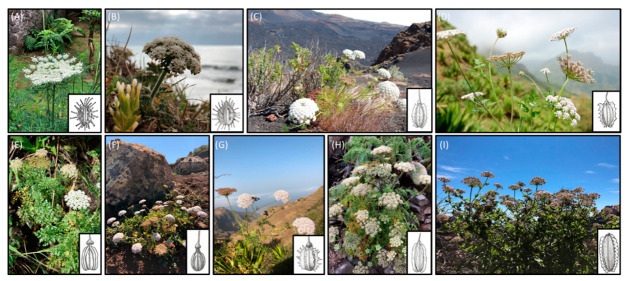
Daucinae subtribe endemic species from Macaronesia and mainland Portugal. Drawings of the fruits of each taxon are displayed. (**A**) *Daucus carota* subsp. *azoricus*; (**B**) *Daucus carota* subsp. *halophilus*; (**C**) *Tornabenea tenuissima*; (**D**) *Tornabenea annua*; (**E**) *Tornabenea ribeirensis*; (**F**) *Tornabenea bischoffii*; (**G**) *Tornabenea insularis*; (**H**) *Monizia edulis*; and (**I**) *Melanoselinum decipiens*.

**Figure 2 plants-10-01954-f002:**
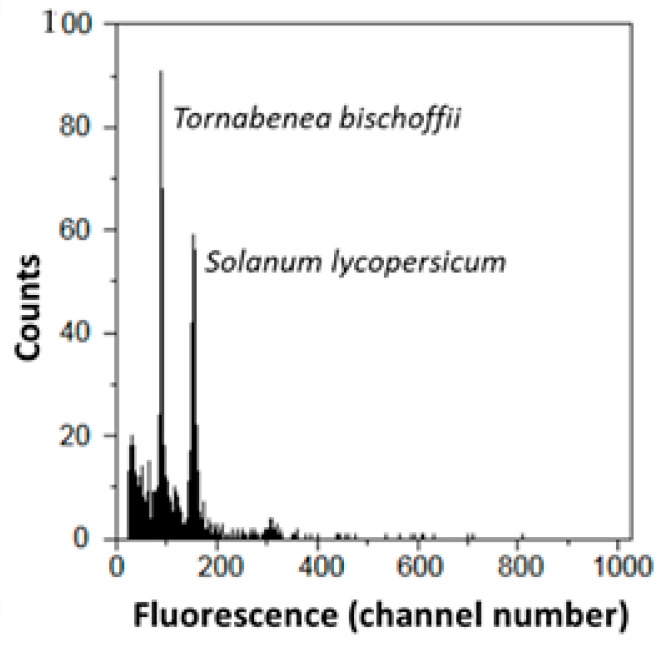
Flow cytometry output: flow cytometric histogram of relative fluorescence intensities from propidium iodide-stained *Tornabenea bischoffii* nuclei using *Solanum lycopersicum* (2C-values = 1.96 pg) as an internal reference standard.

**Figure 3 plants-10-01954-f003:**
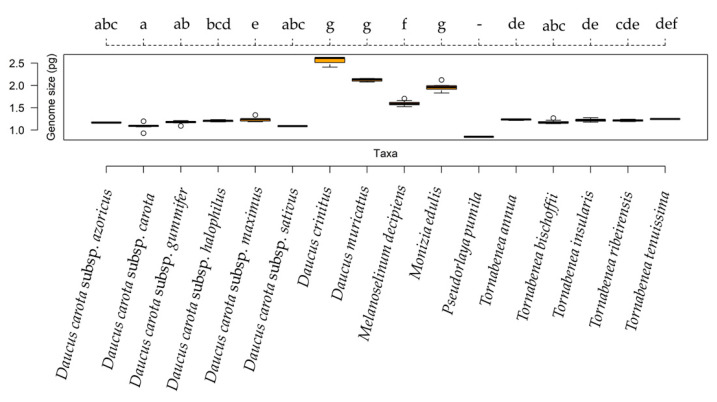
Boxplot diagram showing the genome size variation across the 16 taxa from the Daucinae subtribe. The orange box represents the 25th, 50th (median), and 75th percentiles, while whiskers represent the 10th and 90th percentiles with minimum and maximum observations. The dots represent the outliers. Different letters indicate group of taxa with significant differences.

**Figure 4 plants-10-01954-f004:**
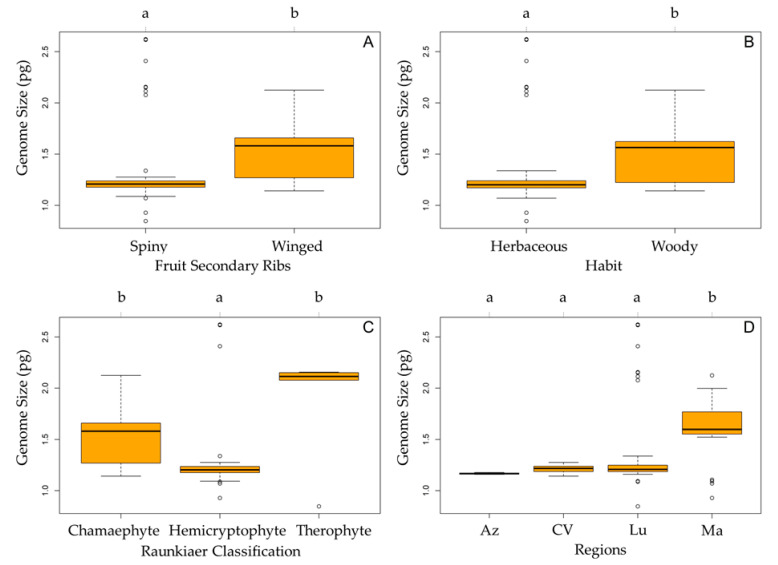
Boxplot diagram showing the genome size variation for the studied Daucinae taxa according to (**A**) morphology of the secondary ribs (winged or spiny), (**B**) habit (herbaceous or woody), (**C**) Raunkiaer classification (chamaephyte, hemicryptophyte, or therophyte), and (**D**) regions (Az, Azores; CV, Cabo Verde; Lu, Portugal mainland; and Ma, Madeira). The orange box represents the 25th, 50th (median), and 75th percentiles, while whiskers represent the 10th and 90th percentiles with minimum and maximum observations. The dots represent the outliers. Different letters indicate significant differences between groups (*p* < 0.05).

**Figure 5 plants-10-01954-f005:**
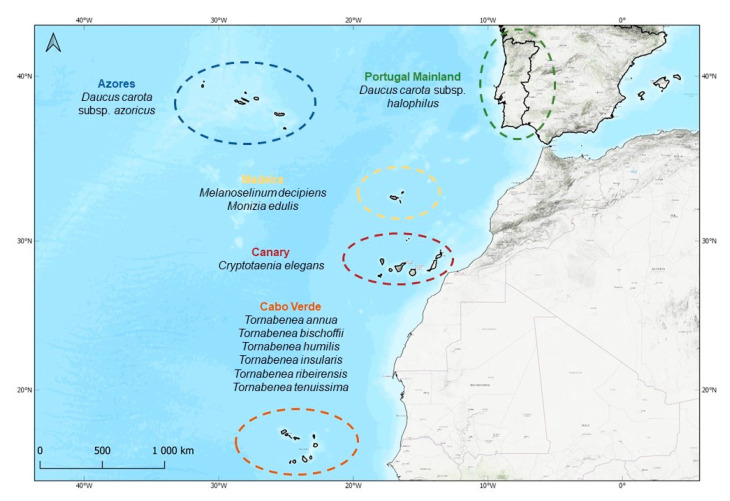
Distribution of the target endemic species in Macaronesia and mainland Portugal.

**Table 1 plants-10-01954-t001:** Geographical distribution, native status in Macaronesia, conservation status (i.e., LC, least concern; NE, not evaluated; DD, data deficient; EN, endangered; and CR, critically endangered (see references in the Material and Methods section), and ecology of the study taxa of Daucinae subtribe in terms of bibliographic revision.

Taxa	Geographical Distribution	Native Status in Macaronesia	Conservation Status	Ecology
*Daucus carota* L. subsp. *azoricus* Franco	Azores	Endemic	LC	Sea rocks, cliffs, pastures, uncultivated or cultivated lands, roadsides; from 0–800 m
*Daucus carota* L. subsp. *carota*	Europa, Asia, Siberia, N. Africa	Native	NE	Ruderal
*Daucus carota* subsp. *gummifer* (Syme) Hook.f.	Europe	-	NE	Coastal cliffs and dunes and uncultivated lands, on sandy soil
*Daucus carota* L. subsp. *halophilus* (Brot.) A. Pujadas	Mainland Portugal	-	DD	Coastal cliffs and coastal upland areas
*Daucus carota* L. subsp. *maximus* (Desf.) Ball	Mediterranean; Asia	Native	NE	Ruderal
*Daucus carota* L. subsp. *sativus* (Hoffm.) Arcang.	Widely cultivated	-	NE	Cultivated
*Daucus crinitus* Desf.	Africa; Europe	-	LC	Grassland, shrubland
*Daucus muricatus* (L.) L.	Africa; Europe	Naturalized	LC	Grassland, shrubland
*Melanoselinum decipiens* (Schrad. & J.C.Wendl.) Hoffm.	Madeira	Endemic	NE	Shady rocks and banks in laurel forests
*Monizia edulis* Lowe	Madeira	Endemic	CR	Mainly found on fissures in cliffs
*Pseudorlaya pumila* (L.) Grande	Mediterranean; Canaries	Native	NE	Marine sands, mainly primary dune
*Tornabenea annua* Bég.	Cabo Verde	Endemic	EN	Southern hygrophyte restricted to montane areas on Santiago
*Tornabenea bischoffii* J.A. Schmidt	Cabo Verde	Endemic	EN	Northern hygrophyte restricted to montane areas on Santo Antão
*Tornabenea insularis* (Parl. Ex Webb) Parl. Ex Webb	Cabo Verde	Endemic	EN	Sub-humid montane areas above 600 m of S. Vicente, S. Nicolau, and Brava
*Tornabenea ribeirensis* Schmidt & Lobin	Cabo Verde	Endemic	CR	Restricted to shady and seasonally damp bottoms of valleys of very few *Ribeiras* in the North of S. Nicolau
*Tornabenea tenuissima* (A. Chev.) A. Hansen & Sunding	Cabo Verde	Endemic	CR	Restricted to sub-humid montane areas; mainly above 1200 m on Fogo

**Table 2 plants-10-01954-t002:** Diagnostic characters of the sampled species of Daucinae subtribe, regarding habit, leaves, inflorescence, bracts, bracteoles, fruits, and secondary ribs in terms of bibliography (see references in the Material and Methods section).

Taxa	Habit	Leaves	Inflorescence	Bracts	Bracteoles	Fruits	Secondary Ribs
*Daucus carota* subsp. *azoricus*	Annual or biennial, up to 70 cm. Herbaceous	Hispid, 2–3 pinnate.	Large terminal umbel, up to 9 cm diameter	6–11 pinnatisect. Filiform lobes.	7–9 simple. Linear lobes.	2–4 mm, cylindrical.	Spiny
*Daucus carota* subsp. *carota*	Perennial, up to 110 cm. Herbaceous.	Basal, 1–3 (4) pinnate oblong to lanceolate, while uper leaves, 1–3 pinnate are linnear to lanceolate.	(1.5) 3–7 (11) cm diameter, becoming strongly contracted in fruit.	7–9 pinnatisect. Sublinear or filiform lobes.	6–9 simple. Linear lobes.	1.8–3.2 × 1–1.8 mm, ellipticals, purplish or light brown.	Spiny
*Daucus carota* subsp. *gummifer*	Perennial, up to 50 cm. Herbaceous.	Basal, (1) 2–3 (4) pinnate. Upper leaves similar to basal ones.	(1.5) 3–6 (10) cm diameter, convex to sub–hemispherical, slightly contracted in fruit.	7–10 shorter than the rays. Linear to lanceolate lobes.	7–9 with a trifid apex. Lanceolate.	1.8–3.0 × 1.3–2.5 mm, oblong to ovoid, brown.	Spiny
*Daucus carota* subsp. *halophilus*	Perennial, up to 25 cm. Herbaceous.	Basal leaves 1–2 (3) pinnate. Upper leaves similar to basal ones, 1–2 pinnate.	(3) 4–12 cm diameter hemispherical and slightly contracted in fruit.	8–10 pinnatisect. Ovoid lobes.	7–8 simple, trifid apex. Ovoid to widely lanceolate.	2–3.5 × 1.5–2.5 mm, ovoid to elliptical, purplish, or brown.	Spiny
*Daucus carota* subsp. *maximus*	Perennial up to 220 cm. Herbaceous.	Basal, (1) 2–3 pinnate ovate to oblong. Upper leaves similar to basal ones 1–2 pinnate.	12–23 cm diameter, becoming strongly contracted in fruit.	10–13 pinnatisect. Linear or filiform lobes.	6–10 simple. Short and linear lobes.	1.5–2.5 × 1–2 mm, ellipsoid–oblong, sometimes subspherical.	Spiny
*Daucus carota* subsp. *sativus*	Perennial, up to 78 cm. Herbaceous.	Basal, 3–4 pinnate, largely petiolate. Upper leaves 2 (3) pinnate.	5–10 cm diameter, slightly convex.	(8) 10–13 pinnatisect. Long and linear lobes.	7–9 simple. Linear to lanceolate lobes.	3–3.5 × 1.2–2.0 mm, oblong, brown.	Spiny
*Daucus crinitus*	Perennial, up to 115 cm.	Basal, 3–4 pinnate, sessile or subsessile segments. Upper leaves similar to basal, 1–3 pinnate.	Convex, does not contract in fruit.	5–10 simple, pinnatisect to trifid. Linear to lanceolate lobes.	5–9 simple. Lobes linear to lanceolate.	4–7 (9) mm, elliptical and sometimes oblong.	Spiny
*Daucus muricatus*	Annual up to 105 cm. Herbaceous.	Basal leaves (2) 3–4 pinnate, hispid. Upper leaves similar but slighty smaller.	Long peduncle. Slightly convex. Sterile central flower absent	(4) 6–10 pinnatisect. Linear or setaceous lobes.	4–9 simple. Linear lobes.	5–8 (10) m, elliptical.	Spiny
*Melanoselinum decipiens*	Tall rosetted perennial monocarpic, up to 3 m. Woody.	Large triangular, up to 60 cm.	50–90 cm diameter, terminal above leaf crown.	10–20, 20–30 mm. Leafy.	As long as the pedicels.	12–14 mm, oblong, pubescent, blackish.	Winged
*Monizia edulis*	Long lived perennial, up to 1 m. Woody.	Yellowish–green, glossy, triangular in out line.	Paniculate, 20–25 rays in each umbel.	Lanceolate or linear, puberulent, fringed at margin.	Lanceolate or linear, puberulent, fringed at margin.	10–14 × 5–7 mm, oblong to ellipsoid, pubescent, pale coloured when ripe.	Winged
*Pseudorlaya pumila*	Annual up to 30 cm. Herbaceous.	2–3 pinnate, hispid.	3–7 unequal rays.	2–5 linear to pinnatisect.	3–5 similar to the bracts but smaller.	(5.5) 7.5–12 × 3.5–10 mm, having the spines in the dorsal and lateral ribs different sizes.	Spiny
*Tornabenea annua*	Annual or biennial, up to 80 cm. Herbaceous.	Up to 35 cm, 2–3 pinnate, 3–6 pairs of pinnae.	More or less flat, up to 7.5 cm diameter.	7–8, entire rarely somewhat bi or trifid. Inconspicuous.	7–8. Inconspicuous.	3.5 mm, strongly compressed dorsally.	Spiny
*Tornabenea bischoffii*	Stout perennial up to 1.5 m. Woody.	35 - 50 cm, 2 (3) pinnate, 7 pairs of pinnae.	Hemispherical, nearly spherical when fruiting, up to 9 cm diameter.	10–15, up to 3 cm long. Pinnately divided.	Trifid, bifid, or entire.	Up to 2 mm long, only slightly compressed dorsally.	Winged
*Tornabenea insularis*	Stout perennial up to 90 cm. Woody.	Up to 30 cm, sub coriaceous to delicate, 1–2 pinnate, 3–6 pairs of pinnae.	Flat to hemispherical, up to 9 cm diameter.	4–13, up to 2.8 cm long, pinnately divided. In fruiting slightly deflexed.	7–9 narrow, trifid, bifid or entire.	2 mm, long elliptical in dorsoventral view.	Spiny
*Tornabenea ribeirensis*	Annual or biennial, up to 80 cm. Herbaceous.	Thin light green, lamina deltoid to obovate in outline, 3–4 pairs of pinnae.	Umbels with up to 25 rays, upward direct bristles, spread, and slightly constricted when fruiting.	3–10 cm long, mainly undivided, rarely one bi or trifid.	Inconspicuous. 2–3 mm long. Undivided.	2.5 mm, compressed dorsally.	Winged
*Tornabenea tenuissima*	Stout perennial up to 1 m high. Woody.	Up to 40 cm, 2–3 pinnate, up to 6 (7) pairs of pinnae. Segments very narrow, filiform.	Hemispherical, nearly spherical when fruiting, up to 9 cm diameter.	Up to 10, 2 cm long. Pinnately divided, segments slender and narrow.	Trifid, bifid, or rarely entire.	Reddish brown, up to 3.9 mm, compressed dorsally.	Spiny

**Table 3 plants-10-01954-t003:** Cytogenomic results obtained for 16 taxa. Average 2C-values in picograms with standard deviation (SD), group (the same letter(s) indicates species within the genus that are not significantly different), sample coefficient of variation as a percentage (CV), previous 2C-value estimates in pictograms based on a previous study [49], and the origin of the collected specimens (Az, Azores; CV, Cabo Verde; Lu, mainland Portugal; Ma, Madeira). The number of chromosomes retrieved from the literature [see references in Section 4.3. Data Collection] is also given.

Taxa	2C-Values ± SD (pg)	Group	Sample CV (%)	Previous 2C-Value (pg)	Chromossome Number	Origin
*Daucus carota* subsp. *azoricus*	1.167 ± 0.024	abc	4.470	1.064	18	Az
*Daucus carota* subsp. *carota*	1.093 ± 0.085	a	4.298	0.989	18	Lu
*Daucus carota* subsp. *gummifer*	1.173 ± 0.036	ab	4.821	1.016	18	Lu
*Daucus carota* subsp. *halophilus*	1.205 ± 0.028	bcd	4.624	1.154	18	Lu
*Daucus carota* subsp. *maximus*	1.244 ± 0.062	e	4.897	0.920	18	Lu
*Daucus carota* subsp. *sativus*	1.087 ± 0.021	abc	4.984	0.960	18	Lu
*Daucus crinitus*	2.544 ± 0.102	g	3.655	2.403	22	Lu
*Daucus muricatus*	2.135 ± 0.040	g	2.314	2.036	20	Lu
*Melanoselinum decipiens*	1.591 ± 0.046	f	3.431	-	22	Ma
*Monizia edulis*	1.940 ± 0.081	g	2.310	-	22	Ma
*Pseudorlaya pumila*	0.847 ± 0.031	-	5.990	-	26	Lu
*Tornabenea annua*	1.235 ± 0.018	de	3.781	-	18	CV
*Tornabenea bischoffii*	1.259 ± 0.211	abc	3.309	-	22	CV
*Tornabenea insularis*	1.223 ± 0.031	de	3.668	-	18	CV
*Tornabenea ribeirensis*	1.214 ± 0.026	cde	4.016	-	-	CV
*Tornabenea tenuissima*	1.248 ± 0.012	def	5.370	-	16	CV

**Table 4 plants-10-01954-t004:** General linear model results for all explanatory variables included in the analysis. The models were ranked by their Akaike’s information criteria (AIC) and adjusted R^2^ values.

General Linear Model	AIC	R^2^
Taxon	−387.044	0.9851364
Region + Raunkiaer classification + habit	63.587	0.3459922
Full	63.587	0.3459922
Region + Raunkiaer classification	63.845	0.3325822
Raunkiaer classification	76.666	0.2203145
Region	79.530	0.2159600
Fruit secondary ribs	89.827	0.1188375
Habit	98.585	0.0548866
Null	103.543	-
Habitat	105.587	0.0004328

## Data Availability

We confirm that all data are original and provided in Tables and Figures within the article.

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
