# Peer review of "Diversity and Cytogenomic Characterization of Wild Carrots in the Macaronesian Islands"

_plants, 2021, doi:10.3390/plants10091954_

Round 1
Reviewer 1 Report
This is an exceedingly interesting and valuable paper which, when combined with previous phylogenetic inference, is very helpful in assessing the evolutionary history of the subtribe Daucinae in the context of Macaronesian geohistory. I have no suggestions for improvement. I am left curious about mechanisms of reproductive isolation in this insect-pollinated lineage.
Author Response
This is an exceedingly interesting and valuable paper which, when combined with previous phylogenetic inference, is very helpful in assessing the evolutionary history of the subtribe Daucinae in the context of Macaronesian geohistory. I have no suggestions for improvement. I am left curious about mechanisms of reproductive isolation in this insect-pollinated lineage.
Response: The authors would like to thank the Reviewer#1 for his very thorough, and positive comments. We have carefully considered the comment “about mechanisms of reproductive isolation in this insect-pollinated lineage” and more details are now provided in the Discussion section. It now reads “The high outcrossing rate (ca. 96%) in wild carrot implies that high frequencies of gene flow may occur among the different subspecies [53]. In addition, pollen of wild carrots could be dispersed by insects over a long distance, and cultivated and wild carrot are fully inter-fertile, often overlap in flowering time, and hybrids may sometimes have high fertility and viability. While gene flow within Daucus carota complex appear extensive, more work is still needed (e.g. integrative genomic and morphological studies) to clearly discriminate population structure within the Daucus carota complex, as well as using additional samples from more diverse geographic origins, to provide future support for recognizing some species and geographically-defined subspecies [19,43]” (Line 330 – 338).
Reviewer 2 Report
The manuscript certainly improves the global knowledge of DNA content for Macaronesian endemics, with special focus on W Mediterranean wild carrots.
The title is properly conceived and well reflects the manuscript contents.
The abstract clearly summarizes the manuscript contents, the same is for the proposed keys words.
The manuscript is overall correctly structured and well written, in a correct English form. Some minor corrections were made directly in the manuscript.
The introduction provides a clear background of the general topic. Methodological approaches are clearly explained and properly structured. Results and discussion are balanced and properly argued, but some more specification are needed to better highlighted the observed variability. The reference list is comprehensive. In Table S2 the syntaxonomic authorships must be provided.
Authors can be find all my observation directly in the manuscript.

Author Response
The manuscript certainly improves the global knowledge of DNA content for Macaronesian endemics, with special focus on W Mediterranean wild carrots.
The title is properly conceived and well reflects the manuscript contents.
The abstract clearly summarizes the manuscript contents, the same is for the proposed keys words.
The manuscript is overall correctly structured and well written, in a correct English form. Some minor corrections were made directly in the manuscript.
The introduction provides a clear background of the general topic. Methodological approaches are clearly explained and properly structured. Results and discussion are balanced and properly argued, but some more specification are needed to better highlight the observed variability. The reference list is comprehensive. In Table S2 the syntaxonomic authorships must be provided.
Authors can be find all my observation directly in the manuscript.
Response: The authors would like to thank the Reviewer#2 for his very thorough, detailed and insightful comments, which have greatly contributed to improving the structure, as well as improving the clarity and readability of our paper.
We would like to highlight that the comments regarding the statistics of the paper were very useful and we reanalyzed our data, namely replacing the ANOVA for Kruskall-Wallis, and we have performed new analyses using the General Linear Model, which we believe that have greatly improved our manuscript. Also two new Figures (Figures 3 and 4) and two new Tables (Table 4 and Table S1) were added in order to reinforce the discussion of the results.
COMMENT 1
In Table S2 the syntaxonomic authorships must be provided.
Response: We agree with the Reviewer and the proposed alterations with information about the syntaxonomic authorships was provided in the revised version of the manuscript (see now Table S3)
Line 92
(…) (Tornabenea insularis) established (…)
Response: Thanks, it has been revised
Line 121 - 122
(…) in the as C–value (…) correlate to (…)
Response: Thanks, it has been revised
Line 141
(…) which occurs in (…)
Response: Thanks, it has been revised
Figure 2 Subtitle
(…) hitogram of (…)
Response: Thanks, it has been revised
Line 270 – 278
The highest mean 2C–values were found in Madeira (1.617 ± 0.183 pg). Comparing the mean 2C–values, it can be observed that the island taxa have lower genome sizes (1.428 ± 0.285 pg) compared to the continental ones (1.544 ± 0.555 pg).
Reviewer Comment: This assertion is not convincing. It counts only considering the overall mean value, but Table 3 actually shows a very variable situation, with continental species showing also the lowest (Pseudorlaya) or various intermediate 2C-values. Authors have to better argument such variability
Response: We recognize that these issues were not sufficiently addressed in the previous version of our study. We now have made changes to the text accordingly. It now reads: “When considering the different regions, the Azores (1.167 ± 0.006 pg) and Cabo Verde (1.238 ± 0.059 pg) showed the lowest mean values, followed by the specimens from mainland Portugal (1.531 ± 0.483 pg) and Madeira (1.617 ± 0.183 pg) (Figure 4D). However, it should be noted that specimens from mainland Portugal included taxa with a wide range of values, from the lowest to the highest, as mentioned above when comparing the results by taxon. As also seen above, the two endemic taxa from Madeira showed intermediate results, while the smallest values for that archipelago were associated with specimens from Daucus carota subsp. carota. This heterogeneity justifies the low adjustment of the respective GLM (Table 4)”
Line 279 - 290
Genome size seems to be correlated to morphological traits. Species with fruits with spiny ribs presented smaller genomes (1.313 ± 0.376 pg) (Figure 3B), and woody species presenting larger genomes (1.521± 0.293 pg) (Figure 3C). However, no correlation was found between life cycle and genome size (Figure 3D)
Reviewer Comment: Again, this statement is not convincing. It is true only considering the overall mean values. Table 3 clearly reveals that D. muricatus and D. crinitus, both with spiny fruits, have the highest 2C-values; also, in fig. 3 the different box plots show the occurrence of too many outliers for each comparison, with exception of fig. 3D. Statistically this condition must be explained.
Response: We agree with this comment and this sentence was rewritten in the new version of the ms. It now reads “With some exceptions, the genomes of herbaceous taxa were smaller than those from woody taxa, with a similar pattern between taxa with spiny and winged fruits. The exceptions are obvious in figures 4A and 4B. Regarding herbaceous taxa, two mainland species appear with very high values, therefore originating a high level of heterogeneity, justifying the low adjustment of the respective GLM (Table 4). Likewise, concerning taxa with spiny fruits, the same taxa from mainland originate a high level of heterogeneity, justifying the low adjustment of the respective GLM (Table 4).
When considering Raunkiaer life-forms (Table S1), therophytes tended to show the highest values and hemicryptophytes the lowest. However, one of the therophytes showed the lowest value and one of the hemicryptophytes showed the highest, therefore originating a high level of heterogeneity (Figure 4C), justifying the low adjustment of the respective GLM (Table 4).”
Line 309 - 318
According to our data the species with the lowest mean 2C–values occur in islands (Madeira not included).
Reviewer comment: I find that using the mean value among taxa is misleading. In fact, based on table 3 and considering the 16 examined taxa, those ones from islands occupy positions from 3 to 10 (except the 7th one) as listed by the largest 2C-value, or from 5 to 10 excluding Madeira.
Authors should better discuss the observed variability, especially with respect to the existing outliers.
Response: We recognize that these issues were not sufficiently addressed in the previous version of our study. We now have made changes to the text accordingly. “In general, the species with the biggest genomes occur in mainland Portugal (Daucus crinitus and D. muricatus) and the insular taxa presented lower mean 2C-values compared to the continental ones. This tendency towards small genomes in islands has been observed in Macaronesia [8,9] and in Hawaiian and Marquesas ar-chipelagos [12]. Even though the Azores and Cabo Verde presented lower mean 2C-values than the mainland Portugal, and this could support the above-mentioned hypothesis, it is important to note that continental regions also presented the taxon with the lowest genome size (i.e., Pseudorlaya pumila) and Madeira presented endemic taxa with larger genome sizes than continental taxa (i.e., Melanoselinum decipiens and Monizia edulis).”
Line 366
Alter-natively, (…)
Response: Thanks, it has been revised
Line 369
Our results showed that taxa with spiny ribs have significantly smaller genomes than taxa with winged secondary ribs.
Response: This sentence was rewritten in the new version of the ms.
Line 393
(…) indicated that (…)
Response: Thanks, it has been revised
Line 397 – 399
Finally, it is require a better understanding of how the intraspecific diversity is changing over time and space and we argue that
Response: This sentence was rewritten in the new version of the ms.
Table 3. The chromosome number of the various taxa was obtained consulting Nowicka et al. [49], Dalgaard [57], Bramwell & Murray [68], Borgen [69], Zizka [70], Grosso et al. [71], and Spooner [72].
Reviewer comment: References should be added in the related column of Tab. 3
Response: Following the Reviewer’s suggestion, the references were incorporated in the Table 3.
Line 488 – 523
Statistical analyses were performed using R software [81]. Descriptive statistics were calculated for each species, namely mean and standard deviation (SD) of the genome size (2C–values, pg). For the data on 2C–values, descriptive analyses were performed using the boxplot statistical algorithm performed by “ggplot2″ package. Comparisons between genome size values and: i) habit (woody/herbaceous); ii) life cycle (annual/perennial); iii) distribution (archipelago/mainland); and iv) morphology of secondary ribs (spiny/winged) were subjected to analysis of variance (ANOVA) and the means were compared using the Tukey test (“TukeyHSD” function; “agricolae” package; [82]). The significance was set to the 95% confidence interval level for all the statistical tests.
Reviewer Comment: Both ANOVA and the Tukey test have as assumption the occurrence of a normal distribution of data in the examined groups. This is not the case of data in this manuscript, as highlighted by the presence of many outliers. Maybe did authors directly compare the overall mean values for each paired group? But in this case, how did they treat the outliers?
Response: We have carefully considered this comment by the Reviewer, and so we would like to highlight that the comments regarding the statistics of the paper were very useful and we reanalysed our data, namely replacing the ANOVA for Kruskall-Wallis, and we have performed new analyses using the General Linear Model (please check the section 4.6. Statistical Analyses), which we believe that have greatly improved our manuscript. We calculated different Gaussian Generalized Linear Models (GLMs), to determine what factors could better explain the observed 2C-values: a Null model was used as a benchmark; an Habitat model (island vs mainland specimens); an Habit model (woody vs herbaceous); a Raunkiaer classification model (chamaephyte vs hemicryptophte vs therophyte); a Fruit type model (specimens with spiny vs wing secondary ribs); a Full model including all the previous factors, as well as other models resulting from its simplification; and a Taxon model, comparing all the included taxa.
Also two new Figures (Figures 3 and 4) and two new Tables (Table 4 and Table S1) were added in order to reinforce the presentation of the results.
Reviewer 3 Report
The manuscript describes the current state of knowledge about wild carrot species in the Macaronesian Islands. Phenotypic descriptions and 2C values are presented.
The genome sizes were quite homogeneous, within and between the analysed species. The conclusion states that further research needs to be done to determine the intraspecific diversity, which can be used for conservation planning, and that cytogenomic analyses can contribute to this. However, I would argue that more detailed phenotypic description of these CWR with an emphasis on traits that could be potentially important for breeding would be more relevant. In addition, analysis of genetic diversity and population structure is also needed to make optimal conservation decisions. It is mentioned in the introduction that ‘A molecular genetics study including a thorough archipelago-wide sampling is currently underway to better clarify the taxonomy of the Azorean Daucus’, but it is unclear if this includes assessment of intra-specific diversity and population structure. Perhaps the concluding paragraph could be modified to include this into the discussion.
Overall, the language quality is good, some minor comments are below, and the manuscript would benefit with a proofreading for minor grammatical corrections.
Minor comments:
Lines 52-53: please clarify this sentence “….with genus-average genome size for islands radiation on the Hawaii and Marquesas archipelagos”. What is meant by islands radiation?
Line 112: “…..variation in the as C–value is….”. Remove the word ‘as’.
Line 113: “….can be correlate…”. Change to ‘correlated’.
Line 334: Maybe ‘relatedness’ would be a better word than ‘proximity’
Lines 388-389: Change “Finally, it is require a better understanding of how the intraspecific diversity is changing over time and space ….” To “Finally, a better understanding of how the intraspecific diversity is changing over time and space is required….”or similar.
Author Response
The manuscript describes the current state of knowledge about wild carrot species in the Macaronesian Islands. Phenotypic descriptions and 2C values are presented.
The genome sizes were quite homogeneous, within and between the analysed species. The conclusion states that further research needs to be done to determine the intraspecific diversity, which can be used for conservation planning, and that cytogenomic analyses can contribute to this. However, I would argue that more detailed phenotypic description of these CWR with an emphasis on traits that could be potentially important for breeding would be more relevant. In addition, analysis of genetic diversity and population structure is also needed to make optimal conservation decisions. It is mentioned in the introduction that ‘A molecular genetics study including a thorough archipelago-wide sampling is currently underway to better clarify the taxonomy of the Azorean Daucus’, but it is unclear if this includes assessment of intra-specific diversity and population structure. Perhaps the concluding paragraph could be modified to include this into the discussion.
Overall, the language quality is good, some minor comments are below, and the manuscript would benefit with a proofreading for minor grammatical corrections.
Response: We would like to thank the Reviewer#3 for the careful and detailed reading of this manuscript and the thoughtful comments. We have carefully considered these comments, and so we have profoundly changed the Discussion and in the Results Section, and reanalysed the data with more species information (Raunkiaer classification: chamaephyte vs hemicryptophte vs therophyte). Two new figures and 2 new Tables were added. Also, more details about the conservation status are provided in Table 1 and in Table 2 about the morphological characters of each study species. Finally, we would like to clarify that the language in the manuscript has been extensively checked and revised so as to improve syntax and the text’s general fluidity. The suggestions put forward, particularly by the Reviewer#2 were all taken into account in the review process.
Comment 1
“It is mentioned in the introduction that ‘A molecular genetics study including a thorough archipelago-wide sampling is currently underway to better clarify the taxonomy of the Azorean Daucus’, but it is unclear if this includes assessment of intra-specific diversity and population structure. Perhaps the concluding paragraph could be modified to include this into the discussion.”
Response: Although we agree with the Reviewer’s concern, we would like to state that assessment of intra-specific diversity and population structure of Daucus carota s.l. in Azores archipelago, is still in progress and there is no accurate information to be provided.
Line 51 – 54
please clarify this sentence “(…) with genus-average genome size for islands radiation on the Hawaii and Marquesas archipelagos”. What is meant by islands radiation?
Response: We agree that this was not clear in the first version, thus we have revised it to clarify the Point. It now reads: “In addition, the number of endemic species per genus was found to be negatively cor-related with genus-average genome size, regarding insular lineages that radiated (i.e. cladogenesis) on the Hawaii and Marquesas archipelagos [12]”
Line 121 - 122
Although the meaning of the variation in the as C–value is still not 113 completely understood, it is known that the variation can be correlate to non-coding DNA, more specifically transposable elements [46].
Response: Corrected in the text. “Although the meaning of the variation in the C-value is still not completely understood, it is known that the variation can be correlated to non-coding DNA, more speifically transposable elements [46].”
Line 397 – 401
Change “Finally, it is require a better understanding of how the intraspecific diversity is changing over time and space ….” To “Finally, a better understanding of how the intraspecific diversity is changing over time and space is required….”or similar.
Response: This sentence was rewritten in the new version of the ms “Finally, a better understanding of how the intraspecific diversity is changing over time and space is required, and we argue that the cytogenomic analyses, can contribute with additional data that can be useful to make informed decisions for the conservation of Plant Genetic Resources in the Macaronesian Islands.”
Round 2
Reviewer 2 Report
The manuscript was significantly improved. Both results and discussions were clarified in their critical parts. New statistical analyses were also correctly applied given more supported information to the data interpretation. All oversights throughout the manuscript were checked.